# Unraveling the genomic complexity of secretion systems in the most virulent *Xanthomonas arboricola* pathovars

Sara Cuesta-Morrondo[1,2], Jerson Garita-Cambronero[3], Jaime Cubero[1]*

1 Instituto Nacional de Investigación y Tecnología Agraria y Alimentaria (INIA), Consejo Superior de Investigaciones Científicas (CSIC), Madrid, Spain, 2 Departamento de Biotecnología-Biología Vegetal, Escuela Técnica Superior de Ingeniería Agronómica, Alimentaria y de Biosistemas, Universidad Politécnica de Madrid, Madrid, Spain, 3 Instituto Tecnológico Agrario de Castilla y León (ITACyL), Castilla y León, Spain

* cubero@inia.csic.es

## Abstract

*Xanthomonas arboricola* pathovars pruni (Xap), juglandis (Xaj), and corylina (Xac) are phytopathogenic bacteria that infect *Prunus* spp., walnut, and hazelnut trees, respectively. In this study, the understanding of the differences among these pathovars was improved with the aim of elucidating their host range and uncovering distinct virulence mechanisms. A comparative genomic analysis was conducted focusing on secretion system clusters across high-quality genomes from two strains of each pathovar. The results revealed that the RaxABC type I secretion system was absent in all analyzed strains. However, the HlyDB type I secretion system was present in both Xap and Xac, with a putative HlyDB effector identified in each Xac strain. Additionally, Xap strains contained a putative PctAB type I secretion system, while only one of the Xac harbored a putative PctAB. Notably, the genomic region surrounding *pctA* and *pctB* lacked *pctP*, suggesting the presence of a novel type I secretion system rather than the canonical PctAB. In contrast, Xaj lacked all the studied type I secretion systems. While the core components of type II and type III secretion systems were highly conserved across strains, significant variation was observed in their substrates. Interestingly, only Xap carried two pathovar-specific type III effectors. Regarding type V secretion systems, complete homologs of EstA, YapH, and XadA were found in all strains, except for one Xac strain, which contained a frameshifted YapH. Additionally, homologs of the XacFhaB/XacFhaC system were found in both Xap strains. However, both Xaj strains and one Xac strain carried an incomplete XacFhaB subunit, while the other Xac strain lacked this system entirely. Finally, analysis of the genomic regions surrounding these secretion system clusters strongly suggests that horizontal gene transfer has played a crucial role in their acquisition, likely contributing to the diversification, emergence and specialization of distinct *X. arboricola* pathovars.

**Data availability statement:** All relevant data are within the manuscript and its Supporting Information files.

**Funding:** This study was supported by PID2021-123600OR-C41 funded MICIU / AEI /10.13039/501100011033/, by "ERDF A way of making Europe" and PRE2019-090846 "ESF Investing in your future". the funders had no role in the study design, data collection and analysis, decision to publish, or preparation of the manuscript.

**Competing interests:** The authors have declared that no competing interests exist.

## Introduction

*Xanthomonas arboricola* is a species of gram-negative rod-shaped bacteria that includes widespread pathogens affecting various plant hosts. This species comprises three main pathovars (pvs.): pruni, juglandis and corylina, which are associated with serious economic losses. These pathovars were first described in North America at the beginning of the 20th century [1–3], and have been detected worldwide in recent decades [4–7].

*Xanthomonas arboricola* pv. pruni (Xap) causes bacterial spot of *Prunus* spp., leading to necrotic spots on leaves, sunken lesions on the fruit and cankers on the twigs and trunk of *Prunus dulcis*, *Prunus domestica*, *Prunus salicina*, *Prunus persica, Prunus armeniaca*, *Prunus avium* and some ornamental trees [8]. In the most severe cases and in some hosts, Xap can cause defoliation, fruit dropping and the death of the tree [9]. *Xanthomonas arboricola* pv. juglandis (Xaj) affects walnut (*Juglans* spp.), causing diseases such as walnut bacterial blight (WBB) which produces necrotic spots on fruit and leaves, brown apical necrosis (BAN) which generates apical necrotic lesions near the blossom end when Xaj is associated with some fungal pathogens [5], and vertical oozing canker (VOC) which causes vertical cankers with brown exudates [10]. Finally, *Xanthomonas arboricola* pv. corylina (Xac) causes bacterial blight in hazelnut (*Corylus* spp.), whose symptoms are spots on leaves, fruits and twigs, and trunk cankers [5]. Furthermore, *X. arboricola* species also comprises other "less successful" pathovars, with lower incidence and expansion, such as pv. celebensis, arracaciae, fragariae and zantedeschiae; and also, several non-pathogenic strains [11,12].

*Xanthomonas* presents diverse elements that participate in disease development, including bacterial secretion systems [13–15]. Bacterial secretion systems are sophisticated molecular machineries responsible for translocating proteins or protein-DNA complexes across the bacterial envelope into eukaryotic or prokaryotic cells or the extracellular space [16]. While type IV and type VI secretion systems are mainly concerned with microbial competition; type I, type II, type III and type V secretion systems play a major role in virulence of *Xanthomonas* spp. [16,17]. In fact, the type III secretion system, responsible for secretion of type III effectors into the plant cell, is considered one of the key virulence determinants in most Gram-negative bacteria. Moreover, type I, II and V secretion systems export other proteins that also mediate infection, such as adhesins or cell wall degrading enzymes [16].

Previous studies have analyzed the presence of some of these secretion systems and their effectors, among other virulence factors, in *Xanthomonas* species [15,16,18–23]. Additionally, our group compared type II and type III secretion systems, type III effectors and other virulence factors among the publicly available draft genomes of some Xap, Xaj and Xac strains, focusing on the mechanisms that are involved in the early stages of the infection [4]. However, an exhaustive study of the structure of the clusters encoding secretion systems in *X. arboricola* had not yet been performed.

Next-generation sequencing (NGS) and advanced bioinformatics tools have improved the quality of the genomes previously analyzed, which are now complete

and closed, enabling more comprehensive genomic analysis and more precise annotation [24]. Mainly, complete and closed genomes have allowed the analysis of gene clusters, which could be challenging with draft genomes due to large clusters occasionally being split across several contigs.

Herein, we have delved into the analysis of the genomes of *X. arboricola* strains from the three most important pathovars within the species. This was accomplished by utilizing the high-quality genomes from a previous study [24], focusing on the secretion systems potentially involved in disease development (type I, II, III and V), and examining their putative involvement in the specific interaction of each pathovar with its respective hosts (*Prunus* spp., *Juglans* spp. and *Corylus* spp.).

## Materials and methods

### Bacterial strains

Six strains of *X. arboricola* were used in this work: *X. arboricola* pv. pruni strains IVIA 2626.1 and CITA 33, isolated from *P. salicina* and *P. dulcis*, respectively; *X. arboricola* pv. juglandis IVIA 1317 and IVIA 2499, isolated from *Juglans regia*; and *X. arboricola* pv. corylina CFBP 1846 and IVIA 3978, isolated from *Corylus avellana*. All of them were isolated in Spain and provided by the Instituto Valenciano de Investigaciones Agrarias (IVIA) or the Centro de Investigación y Tecnología Agroalimentaria de Aragón (CITA), except for CFBP 1846, which was isolated in France and obtained from the International Center for Microbial Resources-Collection Française de Bactéries Phytopathogènes (CIRM-CFBP). Detailed information about the strains can be found in Table 1.

### Genomic comparison of secretion system clusters related to pathogenesis

The complete genomes of *X. arboricola* pv. pruni IVIA 2626.1 (CP076628.1; plasmid pXap41: CP076627.1) and CITA 33 (CP076701.1; plasmid pXap41: CP076702.1), pv. juglandis IVIA 1317 (CP076725.1) and IVIA 2499 (CP076726.1), and pv. corylina CFBP 1846 (CP076619.1) and IVIA 3978 (CP076534.1; plasmid pXac18: CP076535.1), available in GenBank and previously described (24), were used in this work. CDS annotations were manually curated when necessary.

Proteins associated with the secretion systems that may play a role in pathogenesis (type I secretion system (T1SS), type II secretion system (T2SS), type III secretion system (T3SS) and type V secretion system (T5SS)) were identified in the six strains using the blastp algorithm [25]. The reference sequences for these proteins were retrieved from accession numbers indicated in literature, except for the type III effectors (T3Es), for which there were used the accession numbers indicated in the EuroXanth project T3Es database [26,27] and for the Sec and Tat components, which were retrieved from EcoCyc Database [28,29] (S1 Table). Protein hits with a query coverage over 80%, a percentage of identity over 40% and E-value < 0.001 were considered as homologs, criteria that strengthen the analysis by incorporating the E-value, which adds robustness by supporting the inference of functional homology [30]. Web blastp against GenBank non-redundant protein sequences (nr) database was performed when necessary. Web microbial tblastn and blastn searches were performed as needed to assess the conservation of a protein or gene in a pathovar. TXSScan v1.1.3 (contained in

**Table 1. Detailed information on the strains of *X. arboricola* analyzed in the work.**

| Pathovar | Strain | Host | Year of isolation | Geographic origin | GenBank Assembly |
|---|---|---|---|---|---|
| pruni | IVIA 2626.1 | *Prunus salicina* cv. Fortune | 2002 | Don Benito (Badajoz, Spain) | GCA_023375015.1 |
| | CITA 33 | *Prunus dulcis* cv. Guara | 2020 | Calanda (Teruel, Spain) | GCA_023375075.1 |
| juglandis | IVIA 1317 | *Juglans regia* | 1993 | Badajoz (Spain) | GCA_023375035.1 |
| | IVIA 2499 | *Juglans regia* | 2001 | Coria (Cáceres, Spain) | GCA_023375055.1 |
| corylina | CFBP 1846 | *Corylus avellana* | 1975 | Vélines (France) | GCA_023374995.1 |
| | IVIA 3978 | *Corylus avellana* cv. Tonda | 2011 | Vilaplana (Tarragona, Spain) | GCA_023374975.1 |

macsyfinder 2.1.4) was used to validate the blastp results and further search other secretion systems not described in *Xanthomonas* [31] (S2 Table). Genome annotation visualization was performed with Geneious Prime 2024.0.7 (https://www.geneious.com). Gene cluster comparison was executed with Clinker version v0.0.27 [32]. Additional domain analysis was accomplished using InterProScan [33], and signal peptides were predicted with SignalP 5.0 [34], in order to validate low-identity hits and support the putative functional annotation. Transcription activator-like effectors (TALEs) were studied using AnnoTALE [35] and IslandViewer 4 was used to study the presence of genomic islands [36].

## Results

Type I secretion systems (T1SSs) play a role in antimicrobial activity, host cell invasion and adhesion to the host cells and extracellular matrix. The different type I secretion systems (T1SSs) reported in other *Xanthomonas* species have been investigated in the genomes of the Xap, Xaj and Xac strains. RaxABC T1SS from *X. oryzae* pv. oryzae (Xoo) consist of the main structural components: RaxA, which is a periplasmic adaptor protein; RaxB, which is a peptidase-containing ATP-binding cassette (ABC) transporter, and RaxC, which is a TolC-like outer membrane protein. Other accessory proteins include RaxST (a sulfotransferase-like protein), RaxH and RaxR (two-component regulatory system) and RaxX (a ribosomally synthesized, post-translationally modified peptide (RiPP) that mimics plant hormone PSY in *X. oryzae* pv. oryzae) [16,37]. RaxX, RaxST, RaxA and RaxB are codified by the *raxX-raxSTAB* operon [38], which was absent in all the strains under our study. On the other hand, orthologs to other proteins related to RaxABC (RaxC, RaxH and RaxR) were present in all the genomes studied.

The putative HlyDB T1SS from *X. citri* pv. citri str. 306 encloses HlyD, a membrane fusion protein and HlyB, an ABC transporter [39]. Orthologs to the components of the putative HlyDB were found in both Xap and Xac strains but not in Xaj (Fig 1A). Moreover, in each of the Xac strains, a different calcium-binding protein with several RTX domains, which could be a putative T1SS effector (UQQ09977.1 in CFBP 1846 and UQQ15155.1 in IVIA 3978), was found near the HlyDB cluster. The putative effector UQQ09977.1 was next to an IS3 family transposase, while UQQ15155.1 had no mobile genetic elements (MGE) near to it. Furthermore, in both Xap strains, a truncated calcium-binding protein (missing the N-terminus) was identified (KP026_06610 from IVIA 2626.1 and KQR53_14175 from CITA 33). This protein exhibited a C-terminal region, which was almost identical to the ones in the aforementioned putative T1SS effectors from Xac strains and were likely disrupted by an IS3 family transposase.

The putative PctAB T1SS comprises PctA, a putative periplasmic adaptor protein, and PctB, a putative peptidase-containing ABC transporter [37]. The components of a putative PctAB T1SS secretion system may have been identified in IVIA 2626.1, CITA 33 and IVIA 3978 (Fig 1B). The reference sequences PctA and PctB were aligned with two protein sequences in each studied strain. Despite showing a low percentage of identity (Per. Ident < 40% for some of the components), domain comparison of PctA and PctB from *X. oryzae* pv. oryzae PXO99A to the putative PctA and PctB in the studied strains revealed that they shared nearly all the domains. Similarly, domain analysis and blastp results showed comparable findings when aligning RaxA and RaxB reference sequences (ACD57929.1 and ACD57930.1) against the same protein sequences in the studied strains, though with slightly lower percentage identities. For example, in the case of IVIA 2626.1 proteins UQQ05844.1 and UQQ05845.1, the alignment of RaxA and RaxB reference sequences showed 28.68% and 42.17% identity, respectively, compared to 31.87% and 46.48% when using the PctA and PctB reference sequences. Since the presence of RaxA and RaxB was ruled out due to the absence of complete *raxX-raxSTAB* cluster in the strains, the potential existence of a PctAB or another T1SS should be considered. Additionally, the putative T1SS components (putative PctA and PctB) in Xap strains showed low identity when compared to the corresponding PctA and PctB from IVIA 3978 (19.60 and 35.20%, respectively), indicating that a distinct PctAB might be present in Xap strains compared to IVIA 3978. No ortholog to PctA or PctB were found in IVIA 1317, IVIA 2499 or CFBP 1846. The findings obtained through TXSScan regarding T1SSs were consistent with and corroborated the results generated by blastp analysis (S2 Table).

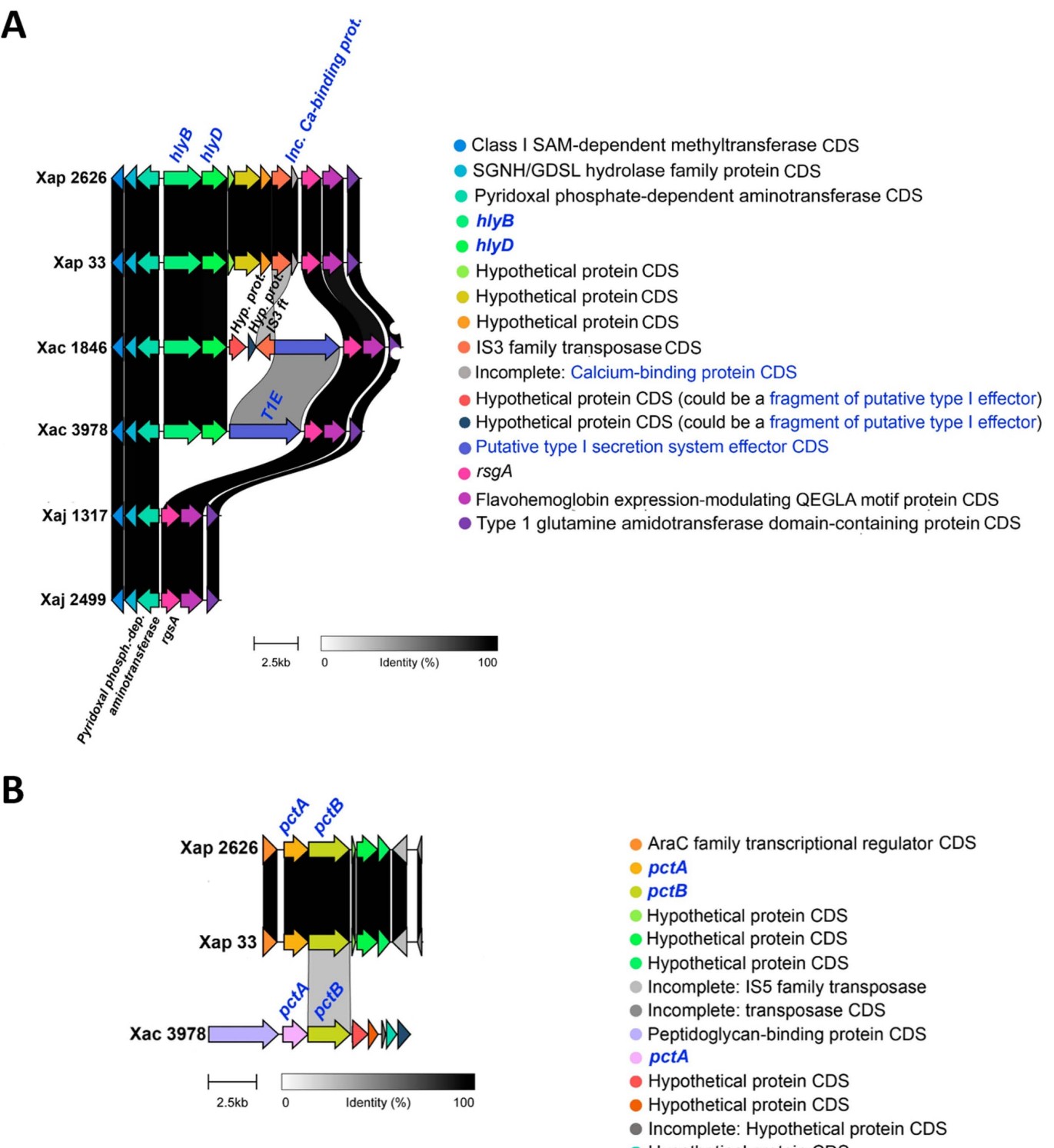

**Fig 1. Type I secretion system (T1SS) clusters in *X. arboricola* strains.** Comparison of genomic regions encoding two T1SS clusters: (A) the *hlyDB* cluster and (B) the putative *pctAB* cluster (or an alternative novel T1SS). Genes corresponding to the secretion systems components and putative effectors are highlighted in blue, while some additional CDS are annotated as follows: Inc. Ca-binding prot. = Incomplete calcium-binding protein; Hyp. prot. = Hypothetical protein; IS3 ft = IS3 family transposase; Pyridoxal phosph.-dep. aminotransferase = Pyridoxal phosphate-dependent aminotransferase. Strains compared include Xap 2626 = IVIA 2626.1; Xap 33 = CITA 33; Xac 1846 = CFBP 1846; Xac 3978 = IVIA 3978; Xaj 1317 = IVIA 1317; Xaj 2499 = IVIA 2499.

The type II secretion system (T2SS) exports folded proteins from the periplasm to the extracellular medium. It depends on a previous transport step from the cytoplasm to the periplasm, which is performed via Sec or Tat pathway [16,40]. All the components of both Sec and Tat secretion pathways were found in all the strains, with almost no differences among them. Herein, the two different T2SS clusters that have been described in *Xanthomonas* were analyzed in the strains studied from the main pathovars of *X. arboricola*. The *xps* cluster codes for proteins XpsE, XpsF, XpsG, XpsH, XpsI, XpsJ, XpsK, XpsL, XpsM, XpsC (previously XpsN) and XpsD, in this exact order; and secretes cellulases, xylanases, lipases, pectate lyases, endoglucanases, polygalacturonases and proteases that degrade the plant cell wall and cause virulence [16,41,42]. The components of this cluster were found in the genome of all the Xap, Xaj and Xac strains studied (Fig 2A). The *xcs* cluster is composed of XcsC to XcsN proteins [16] and has no known function. All the components of *xcs* cluster were present in all the Xaj and Xac genomes, while in Xap strains, *xcsK* was frameshifted (Fig 2B), however, due to the lack of information about the T2SS *xcs* function, the biological implications of this frameshift remain unclear. TXSScan supported the presence of both T2SSs in the strains, although some of the components were not validated in some of them (S2 Table). Moreover, orthologs of eight experimentally validated T2SS substrates from *Xanthomonas citri* pv. citri [43] were searched across all the strains. Orthologs to a cysteine protease (XAC2853), a polygalacturonase (XAC2374), a protease (XAC0795), an endopolygalacturonase (XAC0661) and VirK protein (XAC0435) were found in all the pathovars. In addition, both Xaj and Xac strains also harbored orthologs to an extracellular serine protease (XAC2831).

Comparison of genomic regions encoding two T2SS clusters: (A) the *xps* cluster and (B) the *xcs* cluster. Genes corresponding to the secretion systems components are highlighted in blue. Strains compared include Xap 2626 = IVIA 2626.1; Xap 33 = CITA 33; Xaj 1317 = IVIA 1317; Xaj 2499 = IVIA 2499; Xac 1846 = CFBP 1846; Xac 3978 = IVIA 3978.

The type III secretion system (T3SS) secretes protein effectors (type III effectors, T3Es) targeted to suppress plant immune responses and manipulate host metabolism. Different T3SS are present in plant-interacting bacteria. With a few exceptions, the majority of *Xanthomonas* species that harbor a T3SS carry the Hrp2, whose genes form the hypersensitive response and pathogenicity (*hrp*) cluster [16]. The whole *hrp* cluster, including the components of the different structures that compose the T3SS (the sorting platform, the export apparatus and the needle complex) and the translocation accessory proteins, was found in all the strains. TXSScan also supported the presence of T3SS in all the strains under study (S2 Table). Besides, all the cluster components were almost identical in all the strains (Fig 3).

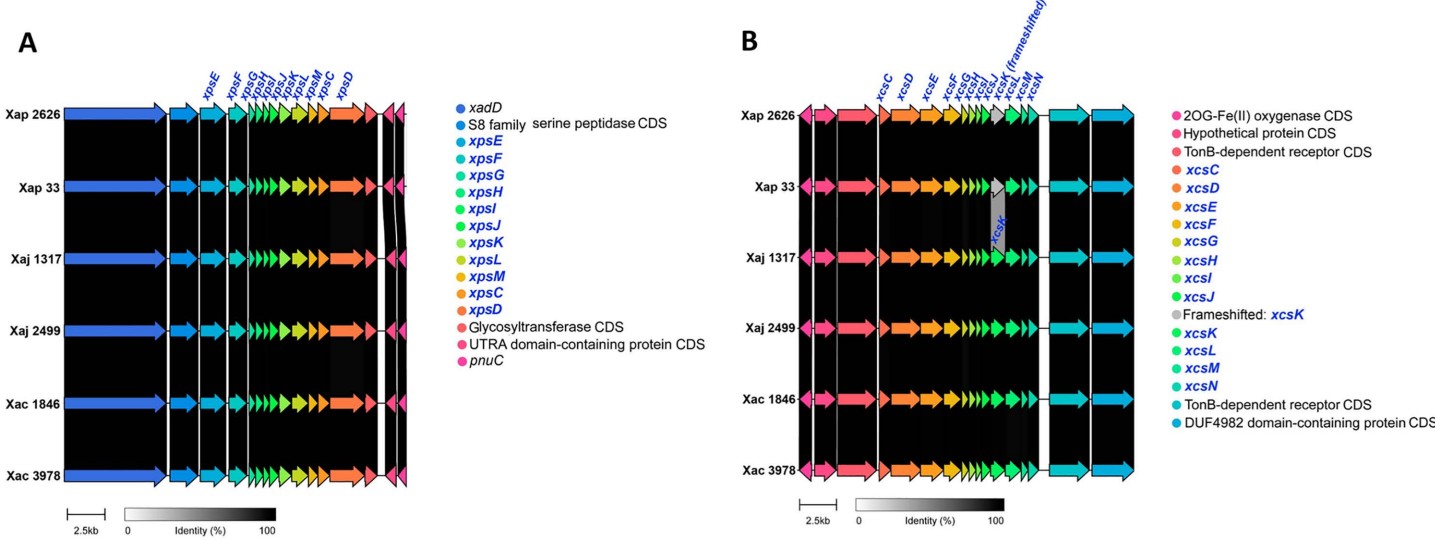

**Fig 2. Type II secretion system (T2SS) clusters in *X. arboricola* strains.**

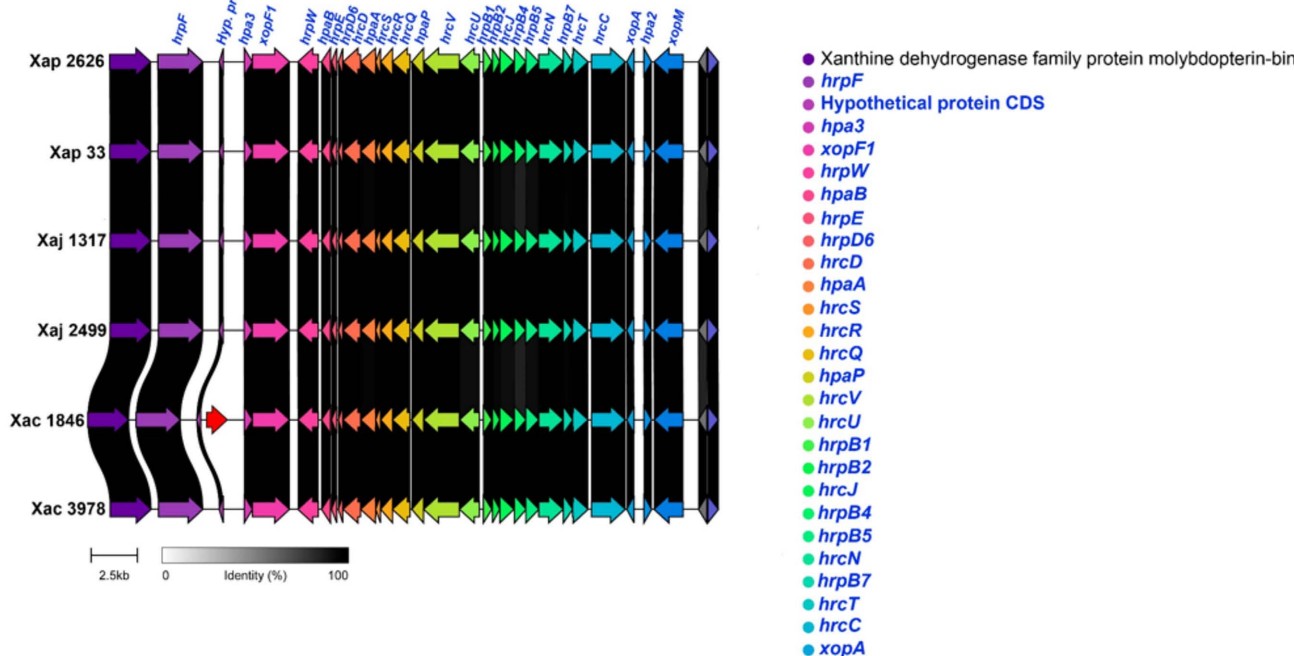

**Fig 3. Type III secretion system (*hrp*) cluster in *X. arboricola* strains.** Comparison of genomic regions encoding the T3SS components and accessory proteins. Genes corresponding to the *hrp* cluster are highlighted in blue, with one CDS annotated as Hyp. prot. = Hypothetical protein. Strains compared include Xap 2626 = IVIA 2626.1; Xap 33 = CITA 33; Xaj 1317 = IVIA 1317; Xaj 2499 = IVIA 2499; Xac 1846 = CFBP 1846; Xac 3978 = IVIA 3978.

The expression of *hrp* cluster and some T3Es is controlled by HrpG and HrpX [43], being both regulators found in all the strains. Other regulatory components that also intervene in the regulation of *hrp* cluster have been described in several *Xanthomonas* species [16], such as HpaS and VemR from *X. campestris* pv. campestris [44,45], which were also found in all the studied genomes. Moreover, several type III effectors were found in the three pathovars, with fifteen of these T3Es shared among the studied strains ([Fig 4]).

However, some effectors were specific to a single pathovar. Both Xap strains contained XopE3, XopAT and XopBA, which were not found in the strains from the other two pathovars, while Xac strains had the unique effector XopE2. Additionally, IVIA 3978 harbored homologs to XopH1, AvrBs1 and AvrBs3 (a transcription activator-like effector, TALE), which were absent in CFBP 1846. In contrast, CFBP 1846 contained XopAG2, which was not present in IVIA 3978. None of the Xaj strains harbored any unique effector.

The type V secretion systems (T5SSs) are responsible for exporting proteins with different functions, including non-fimbrial adhesins, immune evasion and toxin proteins, from the periplasm to the external environment following a step through the Sec pathway. Structurally, T5SS is the simplest of the secretion systems, consisting of a transporter or translocator domain and a passenger domain (the translocated effector), which may even be encoded in the same polypeptide. From the different T5SSs existing subclasses (Va-Vf), only Va (classical autotransporters), Vb (two-partner secretion system, TPS) and Vc (trimeric autotransporter) have been described in *Xanthomonas* [16,46]. One EstA lipase/estearase homolog (Va) was found in all the strains ([Fig 5A]), being its sequence nearly identical across them (99.8%

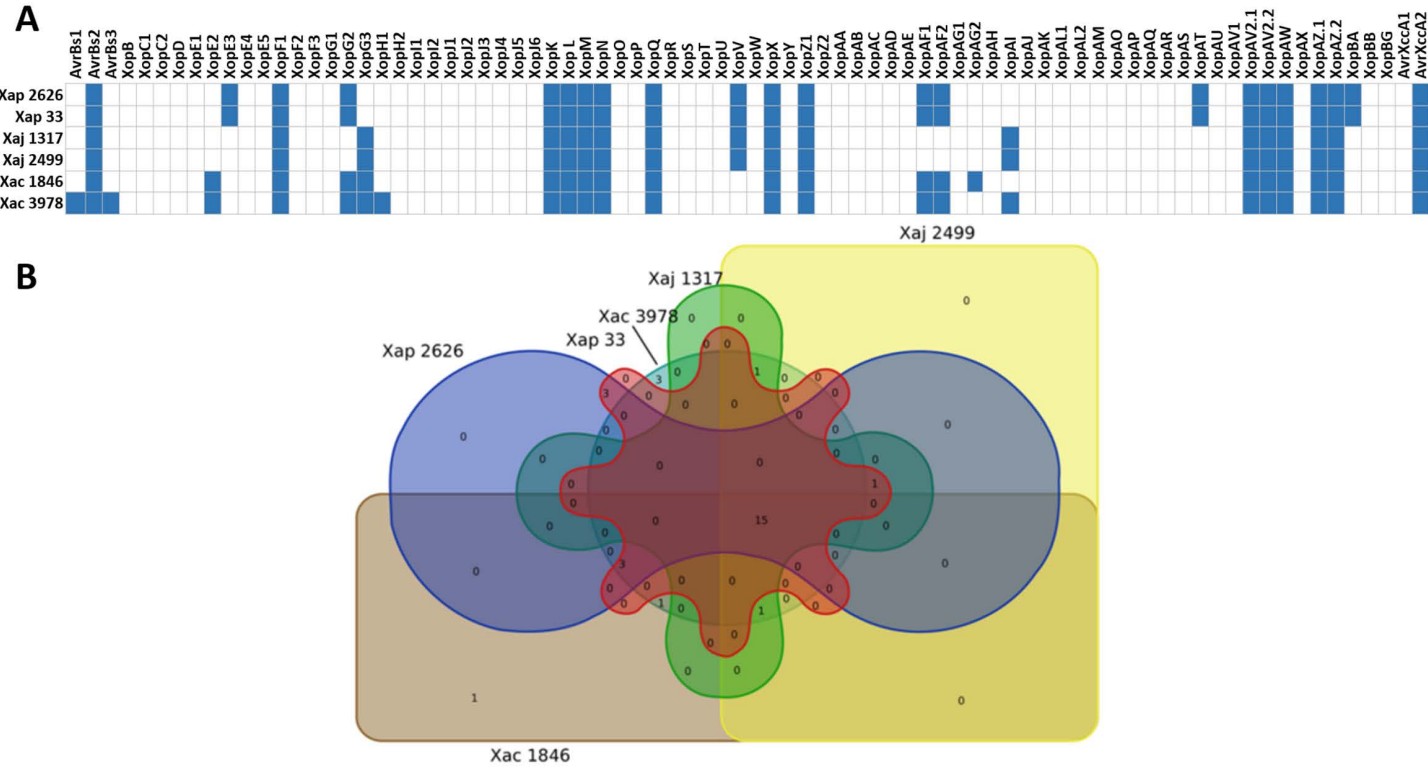

**Fig 4. Type III effectors (T3Es) in *X. arboricola* strains.** (A) Distribution of T3E repertoires identified in the bacterial strains analyzed. (B) Venn diagram showing the shared and unique repertoires of T3Es among the strains compared. Overlapping regions represent effectors common to two or more strains, while non-overlapping regions correspond to strain-specific T3Es. Strains compared include Xap 2626 = IVIA 2626.1; Xap 33 = CITA 33; Xaj 1317 = IVIA 1317; Xaj 2499 = IVIA 2499; Xac 1846 = CFBP 1846; Xac 3978 = IVIA 3978.

identical sites) (S1 and S2 Tables)Furthermore, all the strains had a complete and nearly identical YapH homolog (Va) (97.7% identical sites), except for strain IVIA 3978, where YapH autotransporter homolog exhibited a frameshift mutation (Fig 5B). All the strains, except for IVIA 3978, contained a XacFhaC homolog (Vb). Additionally, a putative XacFhaB was identified next to FhaC in all the FhaC-containing strains, but, in each pathovar, a different version was found. The FhaB-FhaC CDS were located in a region flanked by a tryptophan-rich sensory protein and by the *pgaABCD* operon, which also contained several HGT-related elements, such as transposases and a tRNA (Fig 5C). Regarding *Xanthomonas* adhesin-like protein A (XadA) homologs (Vc), three to four of them were found in each of the studied genomes. The alignment of the XadA reference sequence against all the XadA putative homologs in the studied strains presented relatively low percent identity (from about 32% to 53%) but significant E-value and bit scores. Two XadA homologs were grouped next to each other in all the strains (Fig 5D), separated by a S8 family serine protease. The XadA reference sequence from *X. oryzae* pv. oryzae exhibited around a 50% identity with these two XadA homologs. Moreover, the two homologs only presented about 32% identical sites among them. Other XadA homolog was found in all the genomes (Fig 5E), presenting a 99.5% of identical sites among strains. XadA reference sequence only exhibited about 44% identity compared to this homolog, but it was confirmed by TXSScan. Finally, CFBP 1846 had an additional XadA homolog (UQQ11868.1), confirmed also by TXSScan, to which the reference sequence only showed 32% identity and was located next to a transposase (Fig 5F).

Moreover, three additional T5SSs Va were found with TXSScan (S2 Table, S1 Fig). These would be T5SSs which have not been described yet in *Xanthomonas* and, therefore, were not found with blastp. In summary, although the

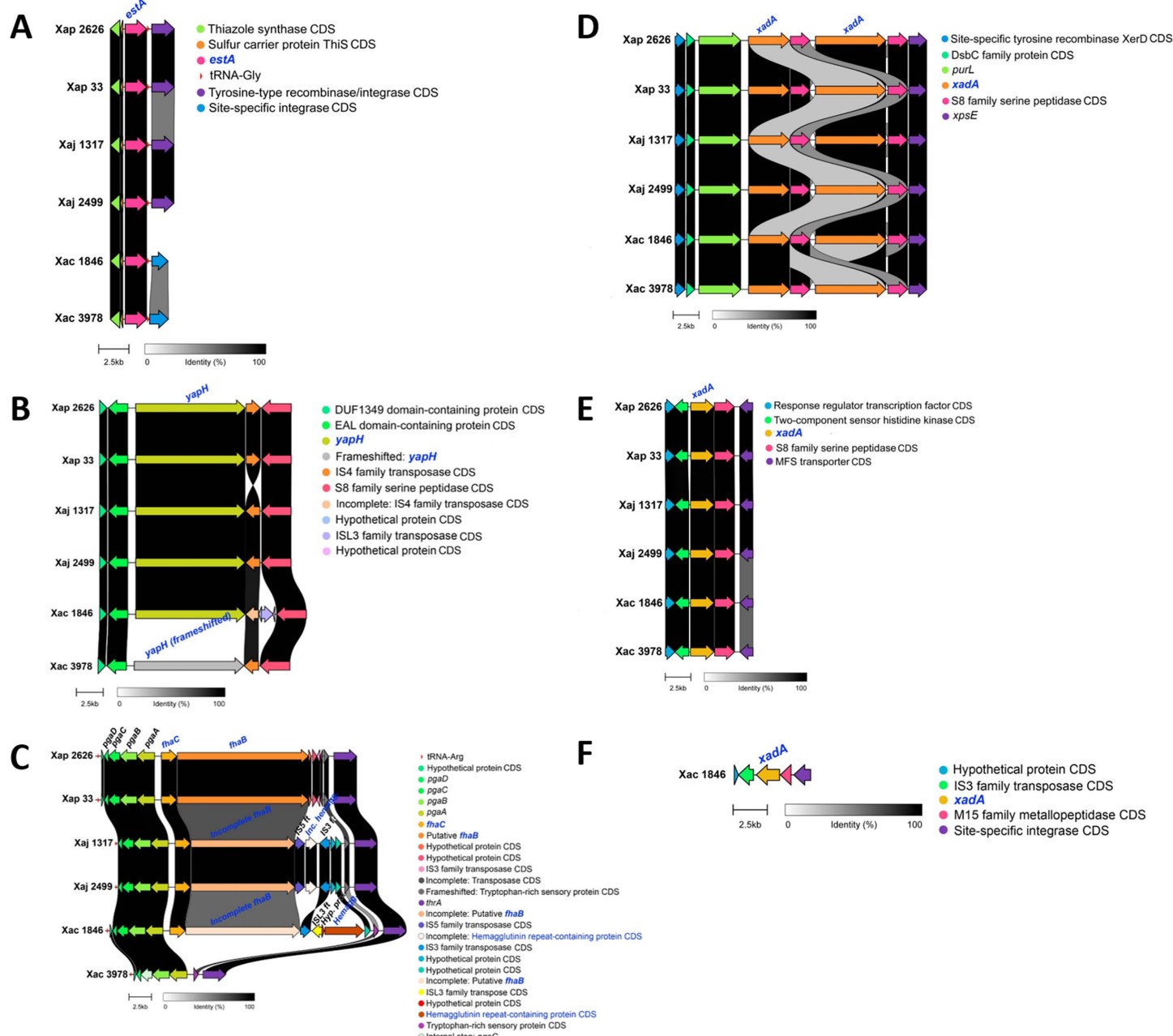

**Fig 5. Type V secretion system (T5SS) clusters in *X. arboricola* strains.** Comparison of genomic regions encoding different T5SS clusters: (A) *estA* homologs, (B) *yapH* homologs, (C) *fhaB/fhaC* homologs, (D) first and second *xadA* homologs, (E) third *xadA* homologs, and (F) fourth *xadA* homolog. Genes corresponding to the secretion system components are highlighted in blue, while some additional CDS are annotated as follows: IS5 ft = IS5 family transposase; Inc. hemagg. = Incomplete hemagglutinin repeat-containing protein; IS3 ft = IS3 family transposase; ISL3 ft = ISL3 family transposase; Hyp. prot. = Hypothetical protein; Hemagg. = Hemagglutinin repeat-containing protein. Strains compared include Xap 2626 = IVIA 2626.1; Xap 33 = CITA 33; Xaj 1317 = IVIA 1317; Xaj 2499 = IVIA 2499; Xac 1846 = CFBP 1846; Xac 3978 = IVIA 3978.

secretion systems in the strains of the three pathovars predominantly share common elements, some differentiating factors exist linked to them (Fig 6). This aspect is particularly notable in the case of the type III secretion system and its effectors (Fig 4).

## Discussion

Comparative genome analysis revealed distinct differences associated with the three pathovars, while also highlighting occasional variations among strains within the same pathovar, suggesting both shared and strain-specific genomic traits.

Different T1SSs were analyzed in the studied genomes. The *raxX-raxSTAB* operon, which codes for some of the structural components of RaxABC T1SS, was absent in all the studied genomes. In *Xanthomonas* species that possess *raxX-raxSTAB* operon, this is flanked by two core genes: *gcvP* and *mfsX* [38]. Both genes were found in all the *X. arboricola* studied strains, with no coding sequences in between. This is consistent with the observations from other authors, who hypothesized that, in the *raxX-raxSTAB* positive species, this cluster had been acquired by general recombination in or beyond *gcvP* and *mfsX* genes after its formation in an ancestor to the lineage containing *X. oryzae*, *X. euvesicatoria* and related species, but not in *X. arboricola* lineage [38]. However, orthologs to RaxC, RaxH and RaxR were found in all the strains. The putative homolog to RaxC was an outer membrane protein from the TolC family and its detection through TXSScan showed that this would participate in the T1SS exports mediated by HlyDB and PctAB, when present [47]. On the other hand, RaxH-RaxR two-component regulatory system is thought to be orthologous to *Pseudomonas* ColS-ColR

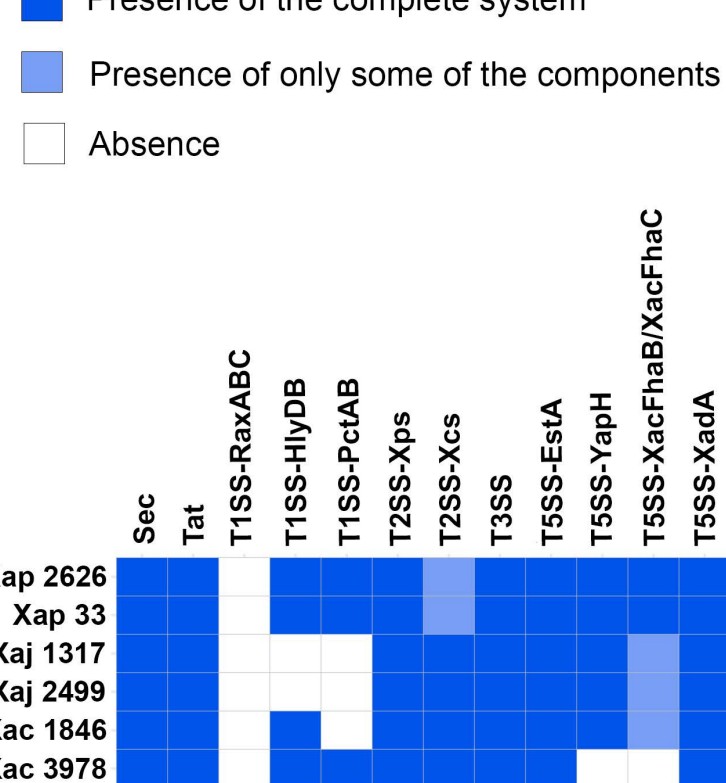

**Fig 6. Comparative genomic analysis of *X. arboricola* pv. *pruni*, *juglandis*, and *corylina* strains.** Distribution of secretion systems associated with pathogenesis in addition to Sec and Tat export pathways. Strains compared include Xap 2626 = IVIA 2626.1; Xap 33 = CITA 33; Xaj 1317 = IVIA 1317; Xaj 2499 = IVIA 2499; Xac 1846 = CFBP 1846; Xac 3978 = IVIA 3978.

system, which responds to $Zn^{2+}$ to control some membrane-related functions, including lipid A remodeling [48]. The role of these regulatory proteins, therefore, could be related to other genes not associated with RaxABC, which may explain their presence in bacteria that lack this T1SS.

The putative HlyDB T1SS cluster from *X. citri* pv. citri (reference sequences: XAC2201 and XAC2202) was not present in Xaj strains, but it was found in all the Xap and Xac genomes analyzed. In general, type I secretion systems export unfolded substrates that include bacteriocins, proteins with nonapeptide repeat in toxins (RTX) motifs, such as proteases and lipases, and large adhesins with several RTX motifs [16]. *X. citri* pv. citri str. 306 harbors two putative RTX effectors (XAC2197 and XAC2198) [39], which were also searched in the analyzed genomes. None of these sequences were found in Xap and Xaj strains. However, in each of the Xac strains, a putative homolog to the toxin XAC2197 was found with a low identity (about 30%) and query coverage ranging from 77 to 83%. InterProScan analysis showed that these sequences were toxins that presented RTX motifs, which bind to the $Ca^{2+}$ present in the extracellular medium, promoting effector folding after secretion [47]. Moreover, these sequences were located near *hlyD* and *hlyB* genes. These two characteristics support their possible role as T1SS effectors (T1Es) [16]. While the putative T1E (UQQ15155.1) in IVIA 3978 had a length of 1,345 amino acids (aa), the one in CFBP 1846 (UQQ09977.1) was 1,230 aa long. Moreover, CFBP 1846 putative T1E was next to an IS3 family transposase and two hypothetical proteins (Fig 1A). These hypothetical proteins, UQQ09979.1 and UQQ09978.1, as well as the UQQ09977.1, shared part of the sequence with UQQ15155.1, which suggests that some kind of reorganization occurred. Furthermore, in the vicinity of the HlyDB cluster of both Xap strains, an incomplete calcium-binding protein was present (KP026_06610 in IVIA 2626.1 and KQR53_14175 in CITA 33), and this was located downstream of three hypothetical proteins and an IS3 family transposase. Both calcium-binding proteins were missing their N-terminus, but their C-terminus aligned with the C-terminus from UQQ15155.1 and UQQ09977.1, suggesting that these calcium-binding proteins could be remnants of T1Es that were possibly interrupted by the insertion of an MGE. Moreover, HlyDB components and putative T1E present in Xap and Xac strains were surrounded by a pyridoxal phosphate-dependent aminotransferase and a ribosome small subunit-dependent GTPase A (*rsgA*) CDS. In the case of IVIA 1317 and IVIA 2499, which lack the putative HlyDB, the pyridoxal phosphate-dependent aminotransferase and the *rgsA* CDS were found, with no other CDS in between. This, added to the presence of insertion sequence (IS) transposases near the HlyDB cluster in some of the putative HlyDB-positive strains, and to the low %GC of some of the CDS in the HlyDB clusters (e.g., UQQ15155.1 shows 53.7% GC) [49], suggests that HlyDB may have been incorporated into some strains through a recombination event, such as the acquisition of a genomic island, only in Xap and Xac studied strains, or its loss in Xaj ones.

It is reported that *X. citri* pv. aurantifolii pathotype B harbored no HlyDB-related genes, while pathotype C harbored HlyDB but not its putative substrates XAC2197 and XAC2198 [39]. The lack of HlyDB-related genes, along with the absence of some key genes of the type IV pilus and the type IV secretion system in *X. citri* pv. aurantifolii pathotypes B and C may explain their lower aggressiveness compared to *X. citri* pv. citri [39]. In our case, we cannot assert that the absence of this cluster in Xaj is linked to differences in strain aggressiveness, partly because we are comparing pathovars of the same species with different hosts. Further studies are needed to confirm this aspect, focusing on clearly establishing the functionality of HlyDB-related genes and their relationship with virulence in *X. arboricola*.

Two putative PctAB T1SS clusters were found, one of them was present in both Xap strains and the other was present in IVIA 3978, but not in CFBP 1846. None of the Xaj strains had a PctAB. InterProScan domain analysis of both PctB proteins showed that they contained a C39 peptidase domain. It was shown that the C39 peptidase domain in RaxB and PctB of *Xanthomonas oryzae* pv. *oryzae* PXO99A harbors a conserved Gln residue and a catalytically essential Cys residue, which may be related to C39 peptidase functionality [37]. These residues were also observed in the PctB proteins from IVIA 2626.1, CITA 33 and IVIA 3978, suggesting these would have proteolytic activity (S2 Fig). Also supporting the similarities between RaxAB and PctAB is the fact that, in both clusters, the periplasmic adaptor protein (RaxA or PctA) is encoded upstream of the ABC transporter (RaxB or PctB, respectively) [37]. However, in the HlyDB cluster of *X. citri*

pv. citri, the ABC transporter HlyB is encoded upstream of the MFP HlyD [39]. These observations agree with the cluster configurations of PctAB and HlyDB clusters analyzed in this work. The role of PctAB as a T1SS is still considered putative, as there is insufficient information about it and its possible substrates [16]. It was demonstrated that *X. oryzae* pv. oryzae PctAB could partially compensate for the absence of RaxAB in the secretion of RaxX. Furthermore, the additional genes within the same genetic region as *pctAB* in Xoo, including *pctP* and an insertion sequence (IS), were identified [37]. While in the PctAB-positive strains analyzed in our study, several MGE-related CDS were present, in the PctAB neighboring areas, no putative *pctP* was found. This could support the previously proposed hypothesis that these clusters would represent distinct T1SSs rather than PctAB. Moreover, the presence of MGE-related CDS near the putative PctAB clusters, as well as the low GC% of some of the genes, could indicate that these clusters may be part of genomic islands [49].

Regarding protein secretion via T2SS, it should be noted that this process requires a prior step of protein export to the periplasm via either the Sec or Tat pathway. The Sec pathway is also essential for protein secretion through T5SS. The Sec secretion pathway has been extensively studied in *Escherichia coli* [50]. The export of unfolded proteins from the cytoplasm to the periplasm mediated by Sec system comprises SecB, a chaperone that recognizes the SecB cleavable signal sequence of the target preprotein and prevents preprotein folding; SecA, which acts as ATPase and also directs the preprotein to the SecYEG channel; and the SecYEG channel itself, the channel through which the protein is exported [40]. The translocon accessory complex SecDF-YajC also facilitates inner membrane translocation [51,52]. Homologs to all these components were found in all the studied strains, which was expected, as this pathway is conserved in bacteria, archaea and eukaryotes [52].

However, as the reference sequences were extracted from *E. coli* proteomes, some of the Sec machinery components showed low query coverage or identity (S1 Table). SecE reference sequence (AAC76955.1) only presented a 63% query coverage compared to SecE homologs in all the studied strains, and SecG reference (AAC76207.1) only showed a 59% query coverage against SecG from Xaj strains. Other components exhibited less than 40% identity. Nonetheless, these putative homologs were confirmed using web blastp against GenBank non-redundant protein sequences (nr) database. This search showed that all the above-mentioned putative homologs showed 100% query coverage and >98% identity with their corresponding *Xanthomonas* Sec components. Regarding its genomic context, SecD and SecF were located next to each other in the three pathovars, which corresponded with previous observations in *E. coli*, where they are, in addition, co-transcribed in a single operon [52]. YajC was also next to SecD and SecF, while the rest of the components of the machinery were scattered along the chromosome.

The Twin-Arginine Translocation (Tat) pathway exports folded proteins from the cytoplasm to the periplasm. The three key components of the Tat pathway in most gram-negative bacteria, TatA, TatB and TatC, were found in the chromosomes of all the studied strains. However, as was the case with some of the Sec machinery components, TatB presented low query coverage, but its identification was confirmed through a blastp search against the nr database. Moreover, TatA, TatB and TatC were located next to each other and may form a single transcriptional unit in the *X. arboricola* analyzed genomes, as described for *X. oryzae* pv. oryzicola [53]. In *E. coli*, the Tat pathway is also composed of TatE, a protein with a sequence related to TatA. However, TatE was not identified in any of the strains studied in this work, which is consistent with previous observations by other authors [53], who did not detect TatE in the genomes of *X. oryzae* pv. oryzicola, *X. oryzae* pv. oryzae, *X. citri* pv. citri and *X. campestris* pv. campestris, but did detect the other three components. The lack of TatE in *X. arboricola* supports their hypothesis of the whole *Xanthomonas* group lacking this component [53]. In addition to their role in the first step of T2SS export, the Tat pathway may be involved in virulence, motility, flagellation, chemotaxis and exopolysaccharide (EPS) production in *X. oryzae* pv. oryzae and pv. oryzicola [53,54].

Regarding the *xps* and *xcs* T2SS, both clusters were found in all the *X. arboricola* studied genomes, which is consistent with previous observations of this species [15,16]. The *xps* T2SS cluster, which plays a role in plant cell wall degradation, was found in all the strains studied, flanked by the same genes in all the cases. The presence of this cluster in all the pathovars agrees with previous observations that detected a high conservation of this cluster across the genus [23]. On

the other hand, the complete *xcs* cluster was found in all the strains except for *xcsK* of Xap, which was frameshifted. In a previous work, after analysis of draft genomes of IVIA 2626.1 and CITA 33, in addition to *X. arboricola* pv. juglandis CFBP 7179 and *X. arboricola* pv. corylina NCCB 100457, among others, only five components of the *xcs* T2SS were identified. This was likely due to the differences in the homology criteria or to the quality of the genomes [20]. Moreover, the presence of the *xcs* cluster only in the *Xanthomonas* species that infect eudicots was reported, which supports the present data, as *Prunus* spp., *Juglans* spp. and *Corylus* spp. are eudicots [23]. However, the role of the whole *xcs* cluster remains unclear, although a possible function in survival under disadvantageous conditions, such as nutrient deprivation, has been suggested for *X. citri* [16]. Furthermore, some authors showed that some *xcs* components partially complemented the loss of homologous *xps* components in *X. vesicatoria (syn. X. campestris* pv. vesicatoria) [41].

Some experimentally validated T2SS substrates were found with blastp. Moreover, these putative T2SS substrates were analyzed with SignalP 5.0. Only the orthologs which contained a signal peptide that targets the protein to the Sec or Tat machinery were considered. A single ortholog to protease XAC0795, polygalacturonase XAC2374 and endopolygalacturonase XAC0661 were found in all the genomes. These two last proteins, the polygalacturonase and the endopolygalacturonase, are cell wall degrading enzymes (CWDE) that hydrolyze pectic substances [43]. Furthermore, two orthologs to VirK protein (XAC0435) were found in all the strains. One ortholog to a cysteine protease (XAC2853) was found in Xap strains, while Xaj and Xac strains harbored two homologs of XAC2853. Moreover, Xaj and Xac strains also presented, respectively, two and one extracellular serine protease (XAC2831) orthologs. The blastp search yielded a third putative homolog to XAC2831 in both Xaj strains (UQQ04656.1 and UQQ00471.1), but this did not contain a signal peptide which could target them to the Sec or Tat machinery and was discarded. Xap putative XAC2831 homolog presented an internal stop codon.

Regarding T3SS, which is considered the main determinant of *Xanthomonas* pathogenesis, all the components of the *hrp* cluster were found in all the Xap, Xaj and Xac studied strains. This included not only the structural subunits but also the accessory translocation proteins that intervene in the translocation, such as HpaA, HrpW, XopA and HpaB [55]. The HrpW found in these *X. arboricola* strains exhibited low query coverage, but InterProScan showed that it harbored pectate lyase domains, which agrees with previous work in HrpW from *Erwinia amylovora* and other phytopathogenic bacteria and rhizobia [56–61]. On the other hand, in our previous work with draft genomes, *hrcU* was not firstly identified in IVIA 2626.1 genome, although this gene was detected by PCR [20]. The present study with high quality genomes confirms the presence of this component. This agrees with other previous research, which showed that *X. arboricola* pvs. pruni, juglandis and corylina harbor a complete T3SS, in contrast to some non-pathogenic *X. arboricola* strains [21,62]. In all the strains under study, the *hrp* cluster presented lower %GC content compared to the rest of the chromosome and was consistently located near tRNA genes, which are typical features of genomic islands acquired through HGT [49]. This has also been observed in other *Xanthomonas*, such as *X. vesicatoria (syn. X. campestris pv vesicatoria)* [55].

The regulators HrpG and HrpX were found in all the strains. HrpG is described as activating the expression of *hrpX* in *Xanthomonas* [43], while HrpX controls expression of other *hrp* genes and some T3Es, usually binding to a plant-inducible promoter (PIP) box [63]. Other genes in the *Xanthomonas* HrpG or HrpX regulons include T2SS substrates, chemotaxis and flagellar biosynthesis genes, transport related proteins and cell wall degrading enzymes [41,64]. On the other hand, HpaS, a sensor histidine kinase which would form a two-component regulatory system along with the response regulator HrpG in *X. campestris* pv. campestris [45], was also present in all the studied strains. Furthermore, HpaR2, which would form another two-component regulatory system with HpaS [45], was also identified in all the genomes encoded next to *hpaS*. Moreover, the response regulator VemR, in addition to regulating other cellular processes, also controls T3SS by enhancing the activation of *hrpX* through HrpG [44]. VemR was also found in all the studied strains.

Fifteen T3Es (homologs to XopL, XopN, XopK, XopX, XopZ1, AvrBs2, XopAW, XopQ, putative AvrXccA2, XopM, XopF1 and two different homologs to XopAZ and XopAV2) were shared between all the pathovars. Some of them, like XopQ, were present in all the studied strains and had also been identified in other bacterial genera, suggesting a broader

role in virulence [55]. On the other hand, all the strains studied in this work presented two homologs for XopAZ and XopAV2, which shared most of their domains, suggesting some possible redundancy. Finally, AvrXccA2 is not considered an effector in a narrow sense, as its function remains unknown [26,65]. IVIA 2626.1 and CITA 33 presented three unique effectors, XopE3, XopAT and XopBA, which were not detected in Xaj and Xac studied strains. A microbial tblastn search against all the complete and draft genomes of *X. arboricola*, as well as its complete plasmids, confirmed that XopE3 and XopAT were only present in Xap genomes and in no other pathovar. Moreover, further blastp searches against all *X. arboricola* strains showed that XopE3 was identified in all the Xap strains (except for Xp10 and Xp4, where it was encoded in the middle of two contigs and, therefore, it appeared truncated), while XopAT was only observed in some of the Xap strains. Furthermore, microbial tblastn search showed that XopBA aligned with Xap, but also with *X. arboricola* pv. arracaciae strain CFBP 7407. On the other hand, CFBP 1846 and IVIA 3978 shared only one unique effector, XopE2. However, microbial tblastn search showed that XopE2 was not exclusively found in Xac, as it was also identified in Xap 15–088 (CP044334.1). Moreover, an incomplete version of XopE2 was found in other Xap strains. Besides, CFBP 1846 harbored XopAG2, which was unique for this strain with respect to the other studied strains, and IVIA 3978 contained three unique effector homologs: XopH1, AvrBs1 and AvrBs3. The AvrBs3 homolog in IVIA 3978 belonged to the Transcription Activator-Like Effectors (TALE) family, whose members bind to the host's DNA, manipulating its transcription [55]. As TALEs are often missed in draft genomes generated with short-read sequencing technologies [66,67], we expected that the complete genomes analyzed in this study would reveal a larger repertoire of TALEs compared to previous analyses of *X. arboricola* draft genomes, in which no TALEs were found [4]. However, the present blastp analysis only yielded the above-mentioned IVIA 3978 AvrBs3 homolog (UQQ17140.1) located in the pXac18 plasmid. Moreover, this result was verified using the AnnoTALE bioinformatic tool. TALEs are characterized as sequences composed of a N-terminal, in which secretion and translocation signals are located, followed by 1.5–33.5 repeats and, finally, a C-terminal containing two types of eukaryotic motifs: a monopartite nuclear localization sequences (NLSs), which localize the effector to the plant cell nucleus, and an acidic activation domain (AD). In each repeat, the positions 12 and 13, named Repeat Variable Diresidues (RVDs), determine the DNA-recognition specificity of TAL effectors [35]. In IVIA 3978 the putative annotated TALE presented 5.5 repeats, with RVD composition HD-HD-NG-NG-HD-NG. However, this TALE is probably nonfunctional, as a minimum of 6.5 repeats are required to induce target gene expression [68]. It can, therefore, be concluded that the absence of a large arsenal of TALEs in the Xap, Xaj and Xac strains was not due to methodological issues related to the short-read sequencing of the previously analyzed assemblies [4] but to the actual scarcity of TALEs in these pathovars. This study definitively confirms the results obtained in the analysis of other *X. arboricola* genomes assembled using less resolutive technologies, such as Illumina MiSeq [18,69]. Hajri et al., (2012) detected a partial sequence of AvrBs3 only in Xac strains using PCR. Herein, AvrBs3 was not detected in any of the Xap and Xaj strains analyzed [70], which supports the lack of TALEs in our studied strains from these pathovars. Finally, Xaj strains did not harbor any unique T3E and presented a slightly smaller repertoire compared to the other two pathovars, which corresponds to previous observations [70]. The lack of plasmids carrying additional T3Es across this pathovar has been noted in literature [18] and could partially explain the lower content in T3Es of these strains.

Several T5SSs homologs were found in the genomes of the six *X. arboricola* strains. All of them were analyzed with SignalP 5.0 and InterProScan to confirm the presence of the Sec signal peptide or the Extented Signal Peptide Region (ESPR). Type Va subgroup secretion systems, also known as classical autotransporters, comprise T5SSs codified in a single polypeptide, where the C-terminal folds into a 12-stranded β-barrel and the N-terminal codifies the passenger domain. These autotransporters translocate lipases, adhesins and proteases [16,46,71]. Two types of Va secretion systems homologs were searched in the studied strains: Est and YapH. All the studied strains presented a homolog to EstA lipase/esterase from *X. oryzae* pv. oryzae, which agrees with the widespread distribution of this protein in *Xanthomonas* spp. [16]. Moreover, all the strains harbored a homologous protein to YapH adhesin from *X. citri* pv. citri, except IVIA 3978, whose YapH homolog was frameshifted. YapH has been related to seed, leaves and abiotic surface adhesion and biofilm

formation in *X. citri* pv. *fuscans* (syn. *X. fuscans* subsp. *fuscans*) [72]. All the YapH homologs present in Xap, Xaj and CFBP 1846 contained the T5SS-exclusive ExtenDed Signal Peptide Region (ESPR) domain, which confirms its actual presence in these strains even if YapH homologs were not found with TXSScan (S2 Table) [73]. This is a 25-aminoacid length N-terminal extension that is found in a minority of T5SSs belonging to type Va, Vb and Vc subgroups. This signal may help to slow down the export of T5SSs proteins, allowing their proper periplasmic transit [46]. Regarding its genomic context (Fig 5B), the YapH homologs were located next to an IS4 family transposase, which may suggest a HGT event.

Type Vb subgroup is also named two-partner secretion system (TPS), as it is the only T5SS in which the passenger domain (TpsA) is synthesized in an independent polypeptide of the translocator domain (TpsB), but both separate proteins are usually encoded in the same operon. Two-partner secretion systems export adhesins, antibacterial toxins related to contact-dependent inhibition (CDI) and some enzymes, among others [16,74]. Except for IVIA 3978, all the studied strains harbored a homolog to FhaC (TpsB) from *X. citri* pv. citri (XacFhaC) and a putative homolog to adhesin FhaB (TpsA, XacFhaB) or its remnants. These FhaC and putative FhaB genes were found next to diverse hypothetical protein and transposase genes and were also surrounded by a tryptophan-rich sensory protein and by the *pgaABCD* operon (*pgaABCD* operon from *E. coli* is equivalent to the *hmsHFRS* of *Yersinia pestis*), implicated in the synthesis of polysaccharide adhesins required for biofilm formation [75,76]. Potnis et al., (2011) also observed the presence of this cluster next to FhaB/FhaC in several *Xanthomonas* species [77]. In IVIA 3978, these loci, the tryptophan-rich sensory protein and the homolog to the *pgaABCD* operon were one next to each other, with no genes in between (Fig 5C). This data suggests that FhaB/FhaC homolog was acquired horizontally in IVIA 2626.1, CITA 33, IVIA 1317, IVIA 2499 and CFBP 1846 or lost in IVIA 3978.

All XacFhaC homologs had a length of 566 aa and exhibited 97.9% identical sites among strains. InterProScan analysis showed that all of them had two POTRA domains, typical of FhaC transporters [16]. With respect to XacFhaB homologs, some of the corresponding CDS had to be manually curated. Both Xap strains presented an identical 4,815 aa putative homolog to XacFhaB, which has been suggested to be involved in biofilm formation, adhesion and virulence [78]. InterProScan analysis showed that both Xap FhaB homologs contained a TPS secretion domain in the N-terminal, which would be recognized and bound by the periplasmic POTRA domains in FhaC [16] and additionally, they included multiple haemagglutinin repeats.

FhaB homologs were detected in NCBI RefSeq annotations of both Xap strains, as they were incomplete in the GenBank annotations used for the rest of the study. Xaj strains harbored a shorter putative FhaB incomplete homolog (3,802 aa) that also contained the TPS secretion domain and several hemagglutinin repeats. This shorter FhaB homolog partially aligned with the N-terminal and central region of XacFhaB (AAM36677.1) and with the FhaB of Xap strains. Moreover, the FhaB homolog in Xaj strains was followed by an IS5 family transposase, and next to this transposase, an incomplete hemagglutinin repeat-containing protein (401 aa) was found, followed by another transposase. Similarly, CFBP 1846 contained an incomplete frameshifted putative FhaB with its corresponding TPS secretion domain and haemagglutinin repeats that partially aligned with XacFhaB. This was followed by two transposases, a hypothetical protein and another hemagglutinin repeat-containing protein (1,417 aa). These observations suggest that the putative XacFhaB homologs in Xaj strains and CFBP 1846 were interrupted and reorganized by the insertion of transposases.

Type Vc subgroup, also known as trimeric autotransporters, comprises self-exported adhesins codified in a single polypeptide [16]. This subgroup includes XadA, which has been shown to be necessary for optimum disease development in *Xanthomonas oryzae* pv. oryzae [79,80]. All the strains harbored three putative XadA homologs, except CFBP 1846, which contained four. Two of these XadA homologs in each strain were found close to each other, being both followed by an S8 family serine protease (Fig 5D). Both pairs of XadA homologs and S8 family serine proteases were not identical to each other. Downstream of these genes, the T2SS *xps* cluster remarked before was found. InterProScan analysis showed that the shorter XadA homolog (1,322 aa) enclosed typical domains of trimeric autotransporter *Yersinia* adhesin A (YadA) and the longer homolog (2,264 aa) contained, in addition to the previously stated, an ESPR. The ESPR extension of the

trimeric autotransporter EmaA has been shown to be important for correct protein folding or assembly into a functional adhesin [81]. Concerning the third XadA homolog in each strain (Fig 5E), this was also upstream of a S8 serine protease. InterProScan analysis showed this XadA homolog, as well as the additional XadA homolog exclusive of CFBP 1846 (UQQ11868.1) contained typical trimeric autotransporter adhesin YadA-like domains, but not ESPR.

TXSScan found three additional T5SSs, all of them belonging to type Va. InterProScan showed that two of them (UQQ05999.1 and UQQ06620.1 in IVIA 2626.1, and their homologs) were peptidases, while the other (UQQ08703.1 in IVIA 2626.1 and their homologs) would have calcium ion binding as its molecular function. The homologs to UQQ06620.1 were encoded near YapH (S1 Fig), while the others were encoded scattered in their respective chromosomes.

To summarize the results on the secretion systems of the three studied pathovars, only minor or no differences were observed in T2SS and T3SS components and regulators. Notably, the presence of the frameshifted *xcsK* in both Xap could impair the functionality of their *xcs* T2SS, potentially compromising their survival under stress conditions [16]. Higher variations in the profile of T2SS substrates and T3Es were detected. Both Xaj strains presented more T2SS substrates than Xap and Xac members. In contrast, Xaj exhibited the lowest number of T3Es, having no unique one. Moreover, IVIA 2626.1 and CITA 33 contained one pathovar-specific T3E, XopE3, which has been selected as target in PCR for diagnosis [21].

With regard to T1SS, all the pathovars lacked RaxABC. Conversely, both Xap and Xac strains contained a putative HlyDB secretion system, but only Xac strains harbored a complete putative HlyDB RTX toxin. On the other hand, both Xap strains and IVIA 3978 contained a putative PctAB or a new type I secretion system. Xaj strains did not harbor any T1SS of the studied classes. Regarding the T5SSs, both Xap and Xaj strains, as well as CFBP 1846, harbored members of the three T5SS subclasses. However, only Xap strains contained a complete XacFhaB homolog.

Moreover, when the search for some of the putative virulence factors was broadened to all the *X. arboricola* genomes deposited in GenBank, it was observed that homologs to these factors were also present in strains belonging to "less successful" pathovars or that do not belong to any pathovar and may not be pathogenic. This would support the hypothesis by Merda et al. (2016) that *X. arboricola* exhibits an epidemic population structure, in which a recombinant network of diverse, non-pathogenic strains and "less successful" pathovars undergoes homologous recombination and horizontal gene transfer [11]. According to the authors, the acquisition of certain virulence factors from this recombinant network under favorable conditions may have led to the emergence of the "successful" pathovars pruni, juglandis and corylina [11]. This could explain why some putative virulence factors were found in one of the pathovars and some non-pathogenic strains, as each of the pathovars would contain an appropriate genetic combination of virulence factors. In this regard, it is expected that it is not a unique factor for a given pathovar but rather the specific combination of several factors that determines the host range.

Furthermore, frequent signs of HGT have been observed in the *X. arboricola* genomes studied in this work. Several of the analyzed secretion systems showed hallmarks of genomic islands (GIs) which are clusters of genes of 10–200 kb that have been acquired by HGT. GIs include conjugative transposons, prophages and integrons, among others, and are characterized by some specific features, including sequence composition bias (e.g., different %GC in the GI compared to the rest of the genome), neighboring tRNA genes and the presence of insertion sequences (ISs) or other MGE-related genes, virulence genes and hypothetical proteins [49]. IslandViewer 4 confirmed that HlyDB and both PctAB putative clusters, as well as some T5SSs, were part of GIs, so they have been horizontally transferred. The detection of HGT signs in the genomes studied in our work demonstrates the involvement of such phenomena in *X. arboricola* pathovars evolution, supporting Assis et al. (2021) observations, which described HGT as a fundamental mechanism in pv. juglandis evolution [18].

Notably, Xap and Xaj strains exhibited minimal differences in their T1SS, T2SS, T3SS, and T5SS components within the same pathovar. In contrast, Xac strains displayed significant variations in these secretion systems and their effectors, likely reflecting the previously observed diversity within this pathovar [82].

## Conclusions

This study provides a deeper understanding of secretion systems in *X. arboricola*. While previous research from our group examined components of some secretion systems in this species [4,20], the present work expands on these findings using high-quality genomes. The previous reliance on draft genomes and highly strict homology criteria led to the omission of certain secretion system components. With this new approach, we were able to analyze not only individual gene or protein sequences but also their genomic structural context. This represents a significant advantage, as visualizing gene clusters allows for the identification of components that might be overlooked under strict homology constraints. Furthermore, cluster analysis provides insights into the surrounding genomic regions, potentially shedding light on horizontal gene transfer events and finding genes that have been frameshifted and that, otherwise, might not be detected.

It is important to acknowledge that this is not an entirely conclusive study and many of the hypotheses presented here require functional validation. Nevertheless, our findings suggest that differences in host range among bacteria of the same species with highly similar genomes, as seen in *X. arboricola*, cannot be attributed to a single gene or a specific set. Instead, they likely result not from individual genes but from a complex combination of factors that collectively determine this specificity. This study of secretion systems establishes a basis for deciphering their functions, notably in the context of host interaction and virulence. Moreover, the present information may eventually be helpful to design new approaches to control and suppress bacterial pathogens in plants.

## Supporting information

**S1 Table. Results of the blastp searches.** 1A. Results of the blastp search for the secretion systems components. 1B. Results of the blastp search for the type III effectors. 1C. Results of the blastp search for the Sec and Tat pathways components.
(XLSX)

**S2 Table. Comparison between blastp and TXSScan results.**
(XLSX)

**S1 Fig. Additional type V secretion systems found with TXSScan. A.** Homologs to UQQ05999.1 from IVIA 2626.1. **B.** Homologs to UQQ08703.1 from IVIA 2626.1. **C.** Homologs to UQQ06620.1 from IVIA 2626.1.
(DOCX)

**S2 Fig. Alignment of cysteine (C) and histidine (H) motifs of the C39 peptidase domain of PctB sequences from IVIA 2626.1, CITA 33, IVIA 3978 and other peptidase-containing ATP-binding cassette transporters described in literature (Luu et al., 2019).** PctB from *Xanthomonas arboricola* strains present the conserved Cys, His and Asp residues (yellow) that are predicted to form the active site and the conserved Gln residue (green) that may form the oxyanion hole. Xap 2626 = IVIA 2626.1, Xap 33 = CITA 33, Xac 3978 = IVIA 3978. Based on Luu et al., 2019.
(DOCX)

## Author contributions

**Conceptualization:** Sara Cuesta-Morrondo, Jerson Garita-Cambronero, Jaime Cubero.

**Formal analysis:** Sara Cuesta-Morrondo, Jerson Garita-Cambronero, Jaime Cubero.

**Funding acquisition:** Jaime Cubero.

**Investigation:** Sara Cuesta-Morrondo, Jerson Garita-Cambronero, Jaime Cubero.

**Methodology:** Sara Cuesta-Morrondo, Jerson Garita-Cambronero.

**Project administration:** Jaime Cubero.

**Resources:** Jerson Garita-Cambronero, Jaime Cubero.

**Supervision:** Jerson Garita-Cambronero, Jaime Cubero.

**Validation:** Jerson Garita-Cambronero, Jaime Cubero.

**Visualization:** Sara Cuesta-Morrondo.

**Writing – original draft:** Sara Cuesta-Morrondo.

**Writing – review & editing:** Sara Cuesta-Morrondo, Jerson Garita-Cambronero, Jaime Cubero.

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
