## [Decision Letter · Decision Letter 0]

30 Jun 2025

PONE-D-25-24971Unraveling the Genomic Complexity of Secretion Systems in the Most Virulent Xanthomonas arboricola pathovarsPLOS ONE

Dear Dr. Cubero,

Thank you for submitting your manuscript to PLOS ONE. After careful consideration, we feel that it has merit but does not fully meet PLOS ONE’s publication criteria as it currently stands. Therefore, we invite you to submit a revised version of the manuscript that addresses the points raised during the review process.

Your manuscript presents an engaging comparative genomics study focused on Xanthomonas and the diversity of its secretion systems. Following peer review, both reviewers concurred that the manuscript would benefit from further refinement, particularly regarding the definition of homologous genes and the application of more robust sequence alignment tools beyond BLAST to enhance validation. I do agree with them.

I would be inclined to consider the manuscript for publication, provided that the authors address the reviewers’ comments. Doing so will significantly strengthen both the clarity of the findings and the overall impact of the manuscript.

We look forward to receiving your revised manuscript.

Kind regards,

Luis D. Alcaraz, Ph.D.

Academic Editor

PLOS ONE

Journal Requirements:

2. Thank you for stating the following financial disclosure: [This study was supported by PID2021-123600OR-C41 funded MICIU / AEI /10.13039/501100011033/, by “ERDF A way of making Europe” and PRE2019-090846 “ESF Investing in your future”.]. 

3. We note that there is identifying data in the Supporting Information file < Supplementary_Table_1>. Due to the inclusion of these potentially identifying data, we have removed this file from your file inventory. Prior to sharing human research participant data, authors should consult with an ethics committee to ensure data are shared in accordance with participant consent and all applicable local laws.

-Location data

Please remove or anonymize all personal information(Names), ensure that the data shared are in accordance with participant consent, and re-upload a fully anonymized data set. Please note that spreadsheet columns with personal information must be removed and not hidden as all hidden columns will appear in the published file.

Reviewers' comments:

Reviewer's Responses to Questions

**Comments to the Author**

1. Is the manuscript technically sound, and do the data support the conclusions?

Reviewer #1: Yes

Reviewer #2: Partly

2. Has the statistical analysis been performed appropriately and rigorously? 

Reviewer #1: Yes

Reviewer #2: N/A

3. Have the authors made all data underlying the findings in their manuscript fully available?

Reviewer #1: Yes

Reviewer #2: Yes

4. Is the manuscript presented in an intelligible fashion and written in standard English?

Reviewer #1: Yes

Reviewer #2: Yes

5. Review Comments to the Author

Reviewer #1: This is a well-conceived and methodologically solid comparative genomics study that explores the diversity of secretion systems (T1SS–T5SS) across six closed genomes representing the most virulent Xanthomonas arboricola pathovars: pruni, juglandis, and corylina. The work contributes meaningfully to the field of plant pathology by offering insights into the genomic architecture underlying host specificity and virulence potential.

The study uses complete (closed) genome assemblies, enabling high-resolution structural comparisons.

Multiple secretion system clusters are characterized comprehensively across multiple strains.

The visual presentation of gene clusters and secretion system comparisons (Figures 1–6) is clear, well-integrated, and supports the narrative effectively.

The manuscript includes an appropriate discussion of potential horizontal gene transfer (HGT) events contributing to genomic diversity and cluster acquisition.

The discovery and discussion of a putative non-canonical PctAB-like T1SS are particularly valuable and appropriately discussed.

Recomendations:

The Methods section states that protein hits were considered homologs when they satisfied three criteria: E-value < 0.001, query coverage >80%, and sequence identity >40%. The authors cite Pearson (2013) [DOI: 10.1002/0471250953.bi0301s42] in support of this threshold, which is appropriate given that the referenced work emphasizes the importance of statistical significance (e.g., E-values and bit scores) over raw percent identity when inferring homology.

However, I recommend that the authors explicitly elaborate on this rationale in the manuscript, especially given the potential concern that a 40% identity threshold may allow the inclusion of distant homologs with uncertain functional conservation, even when high query coverage is used. This is particularly important for secretion system effectors, where minor sequence divergence may translate into major functional differences.

To improve clarity and strengthen the reliability of the results, I suggest the authors:

Briefly summarize Pearson’s argument (e.g., that statistical measures are more reliable than identity percentages alone in homology inference).

Clarify whether domain conservation (e.g., InterProScan) or other functional annotations were used to validate low-identity hits.

Minor Changes

-Typographical edits:

Line 52: "Thes pathovars" → "These pathovars"

Lines 69–70: Consider rephrasing slightly for clarity: “Secretion systems are complex molecular machineries…”

Line 156: “appeared to have been interrupted…” to “were likely disrupted…”

Optional addition:

-A brief sentence in the conclusions and/or discussing how the findings could guide future experimental design (e.g., validation of T1SS function or virulence assays) would help bridge the gap to applied research.

Reviewer #2: In this manuscript, Cuesta-Morrondo and colleagues present a comparative genomic analysis of six previously sequenced Xanthomonas arboricola genomes. Given the well established role of secretion systems in bacterial pathogenicity, the authors focus their analysis on the occurrence and genomic organization of type I, II, III, and V secretion systems across Xa pathovars of agricultural relevance.The authors provide a systematic description of the presence/absnece patterns and genomic arrangement of the aforementioned secretion systems. While primarily descriptive in nature, with the authors limiting their analysis to the identification of gene clusters encoding these secretion systems, the study offers insights that will be of interest to researchers investigating Xanthomonas pathogenicity.

Taking advantage of available high quality genomes, the authors report some interesting findings, including the identification of a TALE effector, which is often overlooked in draft genome analyses due to its repetitive structure. The authors also provide a detailed description of the distribution of distinct type III effectors across the analyzed genomes which may, at least partially, help explain differences in pathogenic profiles and host-pathogen interactions among X. arboricola pathovars.

However, before recommending this manuscript for publication, I have a significant concern regarding the methodological aspect of secretion systems identification.

MAjor comment:

The key methodological limitation of the manuscript is the approach taken to detect secretion systems, as the authors rely only on BLAST searches, mainly against known components of secretion systems from Xanthomonas genomes. While effective for detecting conserved systems, this method may fail to identify divergent or lineage-specific secretion systems, therefore limiting the depth and completeness of the analysis. To address this limitation, I strongly recommend the use of TXSScan (https://github.com/macsy-models/TXSScan), a dedicated tool for the comprehensive identification of bacterial secretion systems. TXSScan integrates HMM profile searche and synteny criteria, offering enhanced sensitivity compared to BLAST based methods alone. Since the primary claim of the study concerns the diversity and organization of secretion systems, reanalysis using TXSScan would strengthen the conclusions and, potentially, could reveal additional systems not detected with the current approach.

Given that TXSScan is relatively simple and straightforward to implement, this reviewer recommend reanalyzing the genomes using TXSScan to cross-validate, and potentially expand,the identification of secretion systems in the studied Xa genomes. This would ensure the completeness and robustness of the findings prior to publication.

Minor comments:

Line 34: "variation was" instead of "variation were."

line 113: Review and correct citation format (Cuesta-Morrondo et al., 2022).

line 189: The authors mention that xcsK is frameshifted…consider claryfing here whether xcsK is essential for T2SS function, especially for readers unfamiliar with the system nomenclature in Xanthomonas.

line 190: Redaction needs to be double checked...consider rephraseing “Moreover, orthologs of eight Xanthomonas citri pv. citri experimentally validated T2SS substrates [42] were” to “Moreover, orthologs of eight experimentally validated T2SS substrates from Xanthomonas citri pv. citri were”

Line 396: Rephrase: "Xac strains harbored two XAC2853 orthologs" to "...harbored two homologs of XAC2853", as the presence of two related genes within the same genome indicates paralogs, not orthologs.

Fig. 1A: Labels are too small to read.

Fig. 3 and 4: These both relate to T3SS and could be merged into a single figure for clarity and consistency

Fig. 5: Again…gene labels are too small, especially in panel 5C

Fig. 6A: Clarify the meaning of the lighter blue boxes in the figure legend. I guess they represent predicted non-functional clusters (due to frameshifts) but this should be explicitly stated at least in the figure legend.

Fig. 6B: This panel conveys essentially the same information as Figure 4A. Consider replacing 4A with 6B to avoid redundancy

6. PLOS authors have the option to publish the peer review history of their article (what does this mean? ). If published, this will include your full peer review and any attached files.

**Do you want your identity to be public for this peer review?** For information about this choice, including consent withdrawal, please see our Privacy Policy .

Reviewer #1: **Yes: ** Israel Muñoz-Velasco

Reviewer #2: No

---

## [Author Response · Author response to Decision Letter 1]

17 Jul 2025

Rebuttal Letter

We are highly grateful to the reviewers for their thorough and valuable suggestions for improving the manuscript, all of them adressed

Editor:

“Your manuscript presents an engaging comparative genomics study focused on Xanthomonas and the diversity of its secretion systems. Following peer review, both reviewers concurred that the manuscript would benefit from further refinement, particularly regarding the definition of homologous genes and the application of more robust sequence alignment tools beyond BLAST to enhance validation. I do agree with them.”

We appreciate the positive comments regarding the article and the work carried out. Concerning the definition of homologous genes, as suggested by one of the reviewers, we have added information in the text about the parameters used to determine homology and emphasized that the identification was based not only on sequence similarity criteria.

Reviewers' comments:

Reviewer #1: This is a well-conceived and methodologically solid comparative genomics study that explores the diversity of secretion systems (T1SS–T5SS) across six closed genomes representing the most virulent Xanthomonas arboricola pathovars: pruni, juglandis, and corylina. The work contributes meaningfully to the field of plant pathology by offering insights into the genomic architecture underlying host specificity and virulence potential.

Thanks for your positive comments.

The Methods section states that protein hits were considered homologs when they satisfied three criteria: E-value < 0.001, query coverage >80%, and sequence identity >40%. The authors cite Pearson (2013) [DOI: 10.1002/0471250953.bi0301s42] in support of this threshold, which is appropriate given that the referenced work emphasizes the importance of statistical significance (e.g., E-values and bit scores) over raw percent identity when inferring homology.

However, I recommend that the authors explicitly elaborate on this rationale in the manuscript, especially given the potential concern that a 40% identity threshold may allow the inclusion of distant homologs with uncertain functional conservation, even when high query coverage is used. This is particularly important for secretion system effectors, where minor sequence divergence may translate into major functional differences.

To improve clarity and strengthen the reliability of the results, I suggest the authors:

Briefly summarize Pearson’s argument (e.g., that statistical measures are more reliable than identity percentages alone in homology inference).

Clarify whether domain conservation (e.g., InterProScan) or other functional annotations were used to validate low-identity hits.

Thank you for the comment that helps clarify the focus of our analysis. We have modified the text according to your suggestion and included the argument used by the authors of the cited manuscript. Indeed, we have encompassed in the text the parameters employed to determine homology and have clarified that the identification relied on more than just sequence similarity criteria.

Minor Changes

-Typographical edits:

Line 52: "Thes pathovars" → "These pathovars"

Ok, corrected.

Lines 69–70: Consider rephrasing slightly for clarity: “Secretion systems are complex molecular machineries…”

Ok, we’ve rephrased it. However, because Type IV and VI secretion systems target the cytoplasm of other bacteria rather than the host or extracellular space, we decided to retain that part of the sentence to reflect this.

Line 156: “appeared to have been interrupted…” to “were likely disrupted…”

Ok, changed.

Optional addition:

-A brief sentence in the conclusions and/or discussing how the findings could guide future experimental design (e.g., validation of T1SS function or virulence assays) would help bridge the gap to applied research.

Thanks for the comment, we have added a sentence regarding. We have added a sentence to this effect at the end of the conclusions, reinforcing the idea that the results may serve as a basis for future functional studies and contribute to the control of phytopathogenic bacteria.

Reviewer #2: In this manuscript, Cuesta-Morrondo and colleagues present a comparative genomic analysis of six previously sequenced Xanthomonas arboricola genomes. Given the well established role of secretion systems in bacterial pathogenicity, the authors focus their analysis on the occurrence and genomic organization of type I, II, III, and V secretion systems across Xa pathovars of agricultural relevance.The authors provide a systematic description of the presence/absnece patterns and genomic arrangement of the aforementioned secretion systems. While primarily descriptive in nature, with the authors limiting their analysis to the identification of gene clusters encoding these secretion systems, the study offers insights that will be of interest to researchers investigating Xanthomonas pathogenicity.

Taking advantage of available high quality genomes, the authors report some interesting findings, including the identification of a TALE effector, which is often overlooked in draft genome analyses due to its repetitive structure. The authors also provide a detailed description of the distribution of distinct type III effectors across the analyzed genomes which may, at least partially, help explain differences in pathogenic profiles and host-pathogen interactions among X. arboricola pathovars.

We sincerely thank you for your positive feedback and for recognizing the significance and relevance of the work presented.

Major comment:

The key methodological limitation of the manuscript is the approach taken to detect secretion systems, as the authors rely only on BLAST searches, mainly against known components of secretion systems from Xanthomonas genomes. While effective for detecting conserved systems, this method may fail to identify divergent or lineage-specific secretion systems, therefore limiting the depth and completeness of the analysis. To address this limitation, I strongly recommend the use of TXSScan (https://github.com/macsy-models/TXSScan), a dedicated tool for the comprehensive identification of bacterial secretion systems. TXSScan integrates HMM profile searche and synteny criteria, offering enhanced sensitivity compared to BLAST based methods alone. Since the primary claim of the study concerns the diversity and organization of secretion systems, reanalysis using TXSScan would strengthen the conclusions and, potentially, could reveal additional systems not detected with the current approach.

Given that TXSScan is relatively simple and straightforward to implement, this reviewer recommend reanalyzing the genomes using TXSScan to cross-validate, and potentially expand,the identification of secretion systems in the studied Xa genomes. This would ensure the completeness and robustness of the findings prior to publication.

Thank you for the suggestion. We have reanalyzed our data using the proposed tool and confirmed the results in most cases. The outcomes obtained with this tool have been cited throughout the text for each of the gene groups studied, thereby completing the work and following the reviewer's recommendation. Moreover, we have included the reference for the tool and an additional supplementary material to compare results between BLAST and TXSScan.

Minor comments:

Line 34: "variation was" instead of "variation were."

Ok, changed

line 113: Review and correct citation format (Cuesta-Morrondo et al., 2022).

Ok, changed

line 189: The authors mention that xcsK is frameshifted…consider claryfing here whether xcsK is essential for T2SS function, especially for readers unfamiliar with the system nomenclature in Xanthomonas.

Thanks for your comment. While there are some studies suggesting possible functions for the Xcs T2SS, its role remains unclear. Therefore, it is not possible to assess if xcsK is essential. We have included some information in the main text regarding this issue

line 190: Redaction needs to be double checked...consider rephraseing “Moreover, orthologs of eight Xanthomonas citri pv. citri experimentally validated T2SS substrates [42] were” to “Moreover, orthologs of eight experimentally validated T2SS substrates from Xanthomonas citri pv. citri were”

Ok, changed. We also revised the whole text.

Line 396: Rephrase: "Xac strains harbored two XAC2853 orthologs" to "...harbored two homologs of XAC2853", as the presence of two related genes within the same genome indicates paralogs, not orthologs.

Ok, changed

Fig. 1A: Labels are too small to read.

The figure has been modified to increase the size of the labels.

Fig. 3 and 4: These both relate to T3SS and could be merged into a single figure for clarity and consistency

By merging 4 with 6b, we preferred to keep 3 and 4 separate for greater clarity for the reader.

Fig. 5: Again…gene labels are too small, especially in panel 5C

Figure was modified to increase label size.

Fig. 6A: Clarify the meaning of the lighter blue boxes in the figure legend. I guess they represent predicted non-functional clusters (due to frameshifts) but this should be explicitly stated at least in the figure legend.

The legend was modified to clarify that they are incomplete secretion systems, containing only some of the components.

Fig. 6B: This panel conveys essentially the same information as Figure 4A. Consider replacing 4A with 6B to avoid redundancy

Part of Figure 6 was merged with Figure 4, but we believe that the summary presented in the new Figure 6 is quite informative for the reader and it should be preserved

---

## [Decision Letter · Decision Letter 1]

28 Aug 2025

PONE-D-25-24971R1Unraveling the Genomic Complexity of Secretion Systems in the Most Virulent Xanthomonas arboricola pathovarsPLOS ONE

Dear Dr. Cubero,

Thank you for submitting your manuscript to PLOS ONE. After careful consideration, we feel that it has merit but does not fully meet PLOS ONE’s publication criteria as it currently stands. Therefore, we invite you to submit a revised version of the manuscript that addresses the points raised during the review process.

The requested revisions have been satisfactorily addressed, as confirmed by both reviewers. Nonetheless, one reviewer and I concur that the manuscript’s clarity would benefit from refining certain statements—particularly within the figure captions—to make them more self-explanatory and comprehensive. You are encouraged to implement these improvements, after which I will be able to accept the article without requiring further revision.

We look forward to receiving your revised manuscript.

Kind regards,

Luis D. Alcaraz, Ph.D.

Academic Editor

PLOS ONE

Journal Requirements:

Additional Editor Comments:

Reviewer #1:

Reviewer #2: 1) The figure legends are, in general, too brief and would benefit from being more descriptive. The legend of Figure 4 is particularly vague: “B. T3Es Venn diagram.” … a Venn diagram of what? Please elaborate (e.g. “Venn diagram showing the shared and unique repertoires of…”).

2) The sentence “TXSScan also supported the T3SS in all the strains under study”....should read "TXSScan also supported the PRESENCE OF the T3SS in all…" ?

Reviewers' comments:

Reviewer's Responses to Questions

**Comments to the Author**

1. If the authors have adequately addressed your comments raised in a previous round of review and you feel that this manuscript is now acceptable for publication, you may indicate that here to bypass the “Comments to the Author” section, enter your conflict of interest statement in the “Confidential to Editor” section, and submit your "Accept" recommendation.

Reviewer #1: All comments have been addressed

Reviewer #2: All comments have been addressed

2. Is the manuscript technically sound, and do the data support the conclusions?

Reviewer #1: Yes

Reviewer #2: Yes

3. Has the statistical analysis been performed appropriately and rigorously? 

Reviewer #1: Yes

Reviewer #2: Yes

4. Have the authors made all data underlying the findings in their manuscript fully available?

Reviewer #1: Yes

Reviewer #2: Yes

5. Is the manuscript presented in an intelligible fashion and written in standard English?

Reviewer #1: Yes

Reviewer #2: Yes

6. Review Comments to the Author

Reviewer #1: The authors have addressed the major points raised during the first round of review. In particular, they have clarified the rationale behind the homology criteria, expanded their methodological framework with additional validation (including TXSScan analyses and domain-based approaches), and refined the manuscript’s language and presentation. The revisions substantially strengthen the methodological rigor and interpretative clarity of the study. The manuscript now provides a clearer explanation of the thresholds used for homology inference, appropriately contextualized, and integrates complementary validation strategies. These additions address the concern regarding the reliability of homology assignments.

Minor textual and stylistic issues noted in the earlier review have been corrected, and the discussion now better situates the findings within the broader context of bacterial pathogenesis and applied plant pathology.

Overall, the revised version of the manuscript is clear and scientifically sound.

Reviewer #2: Cuesta-Morrondo et al. satisfactorily addressed my main concerns, and I have no further major comments on the revised version. I do, however, have a couple of minor observations for authors consideration:

1) The figure legends are, in general, too brief and would benefit from being more descriptive. The legend of Figure 4 is particularly vague: “B. T3Es Venn diagram.” … a Venn diagram of what? Please elaborate (e.g. “Venn diagram showing the shared and unique repertoires of…”).

2) The sentence “TXSScan also supported the T3SS in all the strains under study”....should read "TXSScan also supported the PRESENCE OF the T3SS in all…" ??????

Aside from these minor points, I believe the manuscript meets the publication criteria of PLOS One, and I recommend it for acceptance.

7. PLOS authors have the option to publish the peer review history of their article (what does this mean? ). If published, this will include your full peer review and any attached files.

**Do you want your identity to be public for this peer review?** For information about this choice, including consent withdrawal, please see our Privacy Policy .

Reviewer #1: **Yes: ** Israel Muñoz-Velasco

Reviewer #2: No

---

## [Author Response · Author response to Decision Letter 2]

2 Sep 2025

Rebuttal Letter

Once again we thanks to the reviewers for their interest on the manuscript and their final suggestions for improving it. All of them were addressed.

Editor:

“The requested revisions have been satisfactorily addressed, as confirmed by both reviewers. Nonetheless, one reviewer and I concur that the manuscript’s clarity would benefit from refining certain statements—particularly within the figure captions—to make them more self-explanatory and comprehensive. You are encouraged to implement these improvements, after which I will be able to accept the article without requiring further revision.”

We appreciate again the positive comments regarding the article and the work carried out and we have addressed the final suggestions and corrections from one of the reviewers.

Reviewers' comments:

Reviewer #2:

The figure legends are, in general, too brief and would benefit from being more descriptive. The legend of Figure 4 is particularly vague: “B. T3Es Venn diagram.” … a Venn diagram of what? Please elaborate (e.g. “Venn diagram showing the shared and unique repertoires of…”).

Indeed the reviewer is right. We have modified all the legends according to such comment in this new version

The sentence “TXSScan also supported the T3SS in all the strains under study”....should read "TXSScan also supported the PRESENCE OF the T3SS in all…" ?

Thank you for the comment. The sentence was modified according to reviewer´s comment.

---

## [Editor Report · Decision Letter 2]

5 Sep 2025

Unraveling the Genomic Complexity of Secretion Systems in the Most Virulent Xanthomonas arboricola pathovars

PONE-D-25-24971R2

Dear Dr. Cubero,

We’re pleased to inform you that your manuscript has been judged scientifically suitable for publication and will be formally accepted for publication once it meets all outstanding technical requirements.

Kind regards,

Luis D. Alcaraz, Ph.D.

Academic Editor

PLOS ONE
---

## [Editor Report · Acceptance letter]

PONE-D-25-24971R2

PLOS ONE

Dear Dr. Cubero,

I'm pleased to inform you that your manuscript has been deemed suitable for publication in PLOS ONE. Congratulations! Your manuscript is now being handed over to our production team.

Kind regards,

on behalf of

Prof. Luis D. Alcaraz

Academic Editor

PLOS ONE